# Selection for Reduced Fear of Humans Changes Intra-Specific Social Behavior in Red Junglefowl—Implications for Chicken Domestication

**DOI:** 10.3390/genes13010043

**Published:** 2021-12-24

**Authors:** Johanna Gjøen, Per Jensen

**Affiliations:** AVIAN Behavior Genetics and Physiology Group, IFM Biology, Linköping University, 58183 Linköping, Sweden; johanna.gjoen@liu.se

**Keywords:** aggression, tameness, fowl

## Abstract

The domestic fowl has a different social behavior compared to their ancestor, the red junglefowl. To examine whether selection for tameness has affected their intra-specific social behavior, 32 red junglefowl from two selection lines, one selected for increased tameness and one selected for a high fear of humans for ten generations, were kept in a group of two females and two males each and were observed in a semi-natural undisturbed enclosure. Birds selected for a low fear of humans had more social conflict, and the males from this selection crowed more and were more often observed in low social proximity to others. The high-fear birds spent more time close together with the rest of the group and performed more social, non-aggressive pecking. These results are consistent with known differences between ancestral red junglefowl and domesticated laying hens. Our results show that intra-specific social behavior has been affected as a side-effect of selection for increased tameness. This may have interesting implications for the emergence of the domestication syndrome in chickens.

## 1. Introduction

Domestication has changed several phenotypic traits in animals, including their behavior [1,2]. The history of any species that has gone through domestication can be separated into different phases [3]. The first is the so-called proto-domestication [2], during which animals are loosely associated with humans and evolve an increased tameability. The second phase is where the human-controlled selection starts leading up to the variety of domestic breeds that exist at present. The last phase, where intensive selection for production traits or other desired traits occur, is for most species a relatively recent event, i.e., the last centuries (most livestock) or even decades (e.g., salmon and mink).

One crucial step in the early phases of domestication is that animals are required to tolerate handling by humans and that they can breed in captivity [4]. In fact, Belyaev (1979) hypothesized that selection for tameness was the most important trait to drive the so-called domestic phenotype (also called the domestication syndrome) [5]. This syndrome includes, e.g, changes in body size and composition, brain size and composition, and pigmentation [6]. Furthermore, it includes changes in behavior, such as increased tameness, i.e., a calm exploratory interaction with humans, defined by Price (2002) as “a measure of the extent to which an individual is reluctant to avoid or motivated to approach humans” [2]. Tameness can be measured as a lack of fear towards humans, either as the ability to let a human approach or as the willingness to approach a human themselves [7].

In the present experiment, we focus on the possible role of tameness in the domestication of the most numerous domesticated animal species, the chicken. The ancestor of the domestic chicken is the red junglefowl (*Gallus gallus*), and the domestication process is assumed to have started as early as 5400 BC [8]. Mitochondrial DNA analyses suggest that the domestic chicken has multiple origins in South and Southeast Asia [9]. Previous studies from our group have shown that selection for increased tameness in the ancestral red junglefowl (RJF) changes other traits in a correlated fashion and appears to drive the evolution of a domesticated phenotype, similar to what was found in the famous Farm-Fox studies [10]. For example, RJF selected for increased tameness grow larger, have a higher feed efficiency, lay larger eggs, and have a smaller brain relative to body size, while cerebellum size has increased relative to total brain mass [11,12].

A central aspect of animal domestication is the ability to thrive in a social environment that is quite different compared to that of the wild ancestors. For example, domesticated animals are usually kept in larger and more homogenous groups under more crowded conditions than in the wild. Furthermore, domestic animals need to be able to habituate to regular handling procedures and a variety of social environments. Hence, it is to be expected that intra-specific social behavior may change as part of the domestication syndrome, an aspect that has not rendered much previous research attention. However, a few studies have shown that increased tameness may affect intra-specific social interactions in different species [13,14,15,16,17,18,19].

The domestication-induced changes in the social environment are not the least obvious when it comes to chickens that are often kept in very large groups under highly crowded conditions. For example, the stocking densities for broiler chickens in the EU vary between 33 kg/m^2^ and 42 kg/m^2^ [20]. Failure to adapt to the social environment can lead to chronic stress and reduce the performance in several production traits, such as growth and egg production, and negatively affect the welfare and health of the animal [21]. Selecting animals that have a calm behavior and show reduced fear might thus also alter the social strategies, such as the tendency to freeze or the motivation for social reinstatement [7,22].

The goal of this study was to compare intra-specific social interactions between groups consisting of red junglefowl selected for a high or low fear of humans kept in undisturbed, semi-natural environments. Our main hypothesis was that the differences between the selection lines would resemble those between ancestral red junglefowl and domesticated chickens, where the low-fear line would behave more similar to domesticates.

## 2. Materials and Methods

### 2.1. Animals and Housing 

The animals used were red junglefowl selected for low fear of humans (LF) versus birds selected for a high fear of humans (HF), both from the tenth generation of selection (S10). The first generation (S1) was formed from an outbred group obtained by two generations of outbreeding of two different zoo populations. The birds have been bred and housed with the same experience of humans, and the selection in each generation was based on scores in a fear-of-human test performed on birds when they were 12 weeks old. Briefly, the behavior of each bird was scored during a standardized human approach test on a scale from 1–5, where 1 is the most relaxed and 5 the most fearful. The breeding and selection program has been described in detail elsewhere [15].

For the present experiment, male and female RJF of high-fear and low-fear lines were used (*N* = 32; 8 males and 8 females from each of the selection lines, the age varied between 237–248 days). The test animals were divided into four observation groups, each consisting of two males and two females from the same selection line.

The home pens of the birds had a layout consisting of an indoor pen connected with an outdoor area (each measuring 3 × 3 m). The indoor pens had access to feed, water, perches and nests ad lib on different levels, and a floor cover of wood chips, while the outdoor part contained a ground cover of gravel, a dust bath, and branches for enrichment. Males and females were kept in separate pens. However, they could see each other through the wire mesh since they were kept in neighboring pens at the facility. Hence, no one of the birds were truly strangers to one another. They were moved to the facility at five weeks of age after being hatched and raised under identical conditions at the university hatchery.

### 2.2. Observation Pens

During the behavioral observations, the birds were observed in groups, each consisting of two females and two males from the same selection line. Observations were carried out in a separate lab room, where two semi-natural observational pens were situated. Each pen measured 2.70 m × 2.08 m, height: 1.80 m. The pens were covered with cardboard on one side, so the two groups could not see each other. In the cardboard, there was an opening cut out for the observer to use for observations (161 cm × 30 cm) to minimize the impact of human presence on the behavior of the birds.

Apart from ad libitum access to water, seashells and food, the enclosures were enriched with a dust bath, a nest box, perches, a pool with hay and dried maggots, a cob of sweet corn hanging from the roof, and a section with rocks and plants. The floor was covered with wood shavings. For the recording of social proximity, the pen was divided into four equally sized virtual zones, not visible to the chickens (see Figure 1)

### 2.3. Procedure and Experimental Design

Two days prior to the start of observations, two groups, one from each treatment, were moved from their home pens to the enriched enclosures to habituate to the new environment for about 48 h. Behavioral observations were carried out during the two days following habituation, at 10.00–12.00 a.m. and at 12.30–14:30 p.m., each day.

The recordings were obtained through focal animal sampling, where each individual was observed for 45 s before rotating to the next animal in the group. All occurrences of the behaviors in the ethogram were recorded continuously during the observation time. After each round of recordings of all four animals, the observer turned to the other group and repeated the same procedure. In this way, a total of 160 observation minutes were obtained for each group.

The ethogram comprised social and territorial behaviors, such as agonistic behaviors, vocalization, crowing, social pecking, as well as social aggregation and displacement, as shown in Table 1. Crowing was only recorded in counts, not duration. After the second day of observation, the animals were brought back to their home pens. The enclosures used for the observations were then cleaned and “refurnished” so the next group could move in the following week.

Due to few occurrences, “Give/receive threat”, “Raise hackle threat”, “Chase”, and “Attack” were merged into one category, named “Agonistic behavior” in the analysis.

### 2.4. Statistical Analysis 

The total data set is available in Appendix A. All data were analyzed using the groups as the independent replicates, separated by sex. The data were analyzed using RStudio [23], with generalized linear models, including the effects of sex and selection line as explanatory variables. Normality was tested using the Shapiro–Wilk normality test and visualized with Q-Q plots. Since the data and residuals were not normally distributed, the analyses were carried out using negative binomial distribution and the log link function. The figures are built using a JMP graph builder [24].

## 3. Results

There were generally few incidents of agonistic behavior, but both males and females in the LF groups performed significantly more agonistic behavior than those in the HF groups (effect of selection line: Z = 4.313, *p* < 0.001; Figure 2A). There was no significant effect on the sex on the frequency of agonistic behavior (Z = 0.000, *p* = 1.0000). Furthermore, the frequency of (non-aggressive) social pecking was significantly different between the selection lines, where the HF birds performed more pecking (Z = −4.656, *p* < 0.001), while again there were no significant effects on the sex (Z = −1.442, *p* = 0.149).

The LF birds perched more than the HF (Z = 5.243, *p* < 0.001, Figure 3A). There were no effects on the sex on perching frequency (Z = −0.075, *p* = 0.94). With respect to social perching, i.e., perching together with at least one other bird from the same group (Figure 3B), there was neither any significant effect on the line (Z = 0.103, *p* = 0.918), nor any effect on the sex (Z = −0.273, *p* = 0.785).

The HF birds were more often observed to stay all together in the same zone (Z = −7.670, *p* < 0.001; Figure 4A), but there were no effects on the sex (Z = −0.198, *p* = 0.843). However, the frequency of being alone in a zone was higher in LF birds (Z = 4.110, *p* < 0.001; Figure 4B). This effect was driven mainly by males, and consequently, there was a significant effect on the sex as well (Z = 3.404, *p* < 0.001).

Males from the LF crowed significantly more than those from the HF line (Z = 2.690, *p* = 0.007; Figure 5A). With respect to other vocalizations, there was a tendency for effect on the selection line, where HF birds tended to vocalize more (Z = −1.930, *p* = 0.05; Figure 5B). This effect was mainly driven by females, and consequently, there was a significant effect on sex (Z = −2.362, *p* = 0.02). The frequency of warning calls (Figure 5C) was neither affected by selection line (Z = 0.108, *p* = 0.91) nor by sex (Z = 0.923, *p* = 0.36).

## 4. Discussion

The aim of this study was to examine the possible effects that selection for increased tameness may have on intra-specific social behavior in red junglefowl as a model of the early phases of chicken domestication. We found that birds selected for low fear of humans (LF) had more incidents of social conflict (agonistic interactions), whereas those selected for a high fear of humans (HF) performed more social (non-aggressive) pecking. Furthermore, LF males crowed more and were more often observed on their own within a certain section of the pen, while HF birds spent more time close together with the rest of the group. Hence, our results show that intra-specific social behavior has been affected as a side-effect of selection for increased tameness. This may have interesting implications for the emergence of the domestication syndrome in chickens.

A first and necessary step in any animal domestication process involves a reduction of fearfulness against humans, allowing the animals to thrive and reproduce in captivity [2,25]. The famous Farm-Fox experiment, where foxes were selected for increased tameness only, demonstrated that an array of traits associated with the domestication syndrome evolved as side effects of this selection, e.g., faster early ontogeny, loss of pigmentation, and modified morphology [10]. The fox experiment attempted to study the possible development of a domesticated phenotype in a species that had not been domesticated previously, but here we have chosen to investigate possible correlated effects in the most commonly domesticated species in the world, the chicken. By starting selection with an outbred population of red junglefowl, the ancestor of the chicken [3], we aimed at modelling the earliest phases of chicken domestication, during which increased tameness can be assumed to have played a central role. Since many behavioral and other phenotypic differences between red junglefowl and modern domesticates have previously been studied in detail, we can effectively determine the extent to which similar domestication effects will develop as a result of increased tameness [17,26]. Previously, we have shown that red junglefowl selected for increased tameness lay larger eggs, have about 10% larger offspring at hatching (weight) and also grow larger, and have an increased feed efficiency and modified brain size and structure [25]. These changes are all in line with how present-day chickens differ from ancestral red junglefowl and are therefore consistent with the idea that increased tameness may drive the domestication syndrome.

In the present study, we extended the previous studies to focus on intra-specific social behavior. Previously, we have found that social reinstatement tendency in young chicks was only moderately affected after five generations of selection [27] in spite of the fact that there is a significant genetic correlation between social reinstatement and tameness [15]. In generation six, there was a certain effect of selection for tameness on agonistic and sexual behavior in adult birds [13]. Here, we therefore performed a systematic study of the undisturbed intra-specific social behavior in groups of selected red junglefowl housed under semi-natural conditions, where they could express a variety of behaviors.

Low-fear (LF) birds showed less social cohesiveness, and the males from this line were more often observed alone in their part of the pen. Furthermore, there was a higher frequency of agonistic interactions and a lower frequency of non-aggressive pecking in LF birds, both of which may be contributing reasons for why birds from this line dispersed more. Since the sexes were housed seperate with both LF and HF birds together in their home pen when they were not in the experiment, and since they had two days of habituation before the observations started, we argue that the agonistic behavior was not due to the birds experiencing each other as strangers. Domesticated white leghorn chickens tend to show more aggression after regrouping and tend to disperse more as a result of increased foraging activity, so the present results are largely consistent with previously known effects of domestication [28,29,30]. The present results may also be related to the fact that LF birds may have experienced more successful social encounters in their home pens, which they shared with the HF birds of the same sex. As mentioned above, LF birds are larger than HF birds and, therefore, more likely to win aggressive encounters [31,32,33], as also found in previous studies of the same selection lines [14]. Accordingly, they may have been more predisposed for aggressive behavior in the newly composed observations groups [34].

LF birds also perched more than HF birds, but during perching, there was no effect of selection on cohesiveness. Red junglefowl males tend to perch more frequently than domesticated white leghorn males [30], which is opposite to the tendency observed in the present study. Perching is an anti-predator strategy, and it is possible that the motivation to perch during daytime is affected by the perch placement and the structure of the environment.

With respect to vocalizations, the clearest effect was a higher frequency of crowing in LF males. This study did not record the duration of crowing, but that could be interesting to research in future studies. Crowing is a territorial announcement also used in intra-specific communication to establish social dominance [35], and dominant males crow significantly more than subordinate males [36,37]. This is consistent with the higher frequency of agonistic interactions in the LF birds, indicating more social competition in these groups. Previously, we have found that LF birds are dominant over HF birds when competing over limited resources [38], and this may possibly be related to a higher feeding motivation since the metabolism and feed efficiency has been altered by the selection procedure [14].

Red junglefowl is a highly social bird species, often spending their entire lives in stable family groups [39]. During domestication, chickens have been selected for the ability to thrive and reproduce in large, dynamic groups, often consisting of birds of the same sex and age [40]. This large alteration in the social living conditions places new demands on the ability to interact and communicate, so it is not surprising that domesticated chickens show an array of differences in intra-specific social behavior compared to their ancestors. The present study suggests that some of these modifications to the social behavior may arise as correlated responses to the reduced fear of humans that is a prerequisite for successful initial domestication.

## 5. Conclusions

In conclusion, birds selected for reduced fear of humans performed more intra-specific agonistic behaviors, crowed more, perched more, and were more often observed on their own within a certain section of the pen. These results suggest that intra-specific social behavior has been affected as a side-effect of selection for increased tameness. It is, therefore, possible that modifications of social behavior in chickens emerged early during domestication as a result of increased tameness.

## Figures and Tables

**Figure 1 genes-13-00043-f001:**
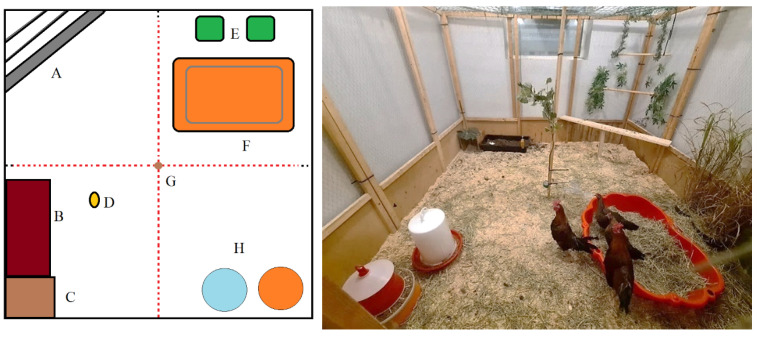
Experimental setup of the enriched observation enclosures. (A) Perches, (B) Dust bath, (C) Nest box, (D) Corncob, (E) Plants, giant Miscanthus, (F) Pool with hay and worms, (G) Pole with seashell supply, (H) Food and water. The dotted line shows the borders between four equally sized virtual zones, not visible to the chickens. The picture to the right shows the view of the pen from the perspective of the observer.

**Figure 2 genes-13-00043-f002:**
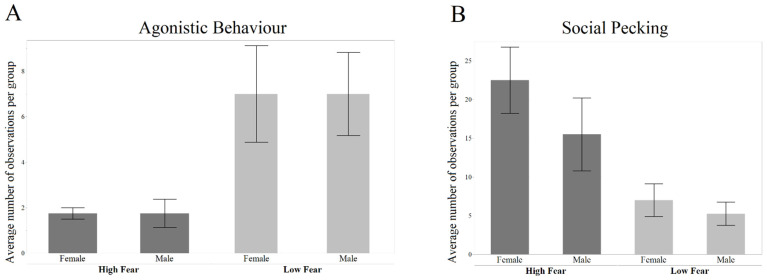
Frequency of agonistic behavior and non-aggressive pecking (mean ± SE). (**A**) Average number of observations per group of agonistic behavior. (**B**) Number of observations of non-aggressive social pecking.

**Figure 3 genes-13-00043-f003:**
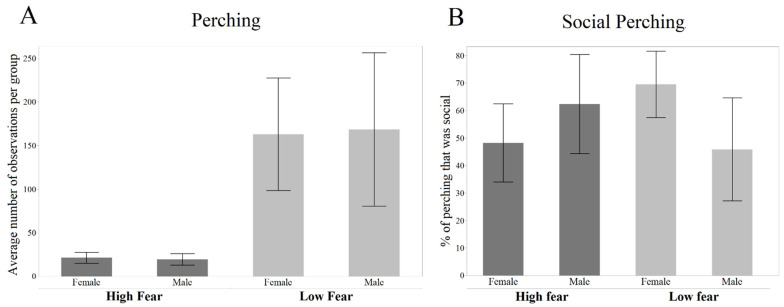
Frequency of perching in total and in a social context (mean ± SE). (**A**) Average number of observations per group of perching. (**B**) Number of observations of a bird perching together with at least one other bird.

**Figure 4 genes-13-00043-f004:**
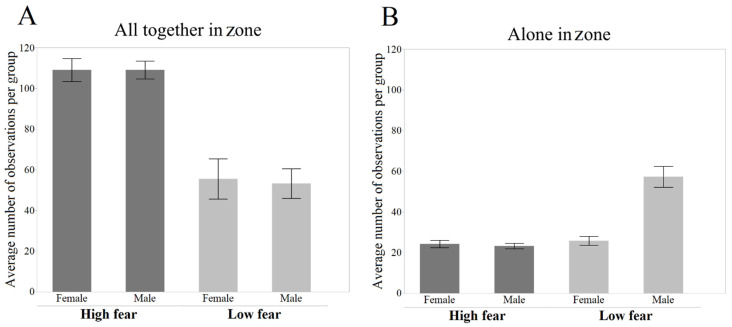
Observations of social proximity in the two selection lines (mean ± SE). (**A**) The number of observations of all four birds staying together in the same zone. (**B**) The number of observations of one bird staying on its own in a zone.

**Figure 5 genes-13-00043-f005:**
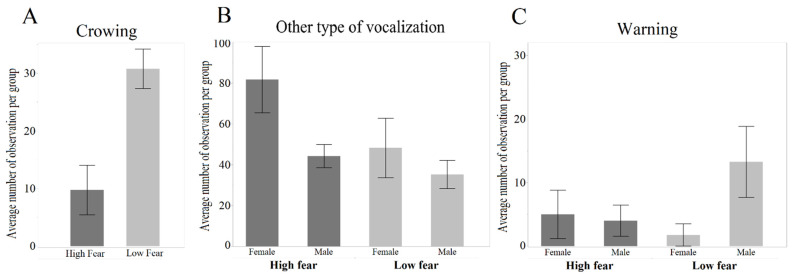
Frequencies of vocalizations in the two selection lines, separated by sex (mean ± SE). (**A**) Average number of Crowing (males only) per group. (**B**) Average number of other vocalizations. (**C**) Average number of warning calls.

**Table 1 genes-13-00043-t001:** The ethogram used in the behavior recordings.

Behavior	Definition
Social pecking	Non-aggressive pecking or manipulating gently at other
Warning	Excited and exaggerated repeated “ka-ka-ka-kaaa”
Crow	Cockerel crowing
Other vocalization	Any unspecified vocalization, not cockerel crowing or warning
Give/receive threat	Bird directs beak towards another bird with head high, the other bird moving away by walking, running, jumping, flying
Raise Hackle threat	Body horizontal or in pecking position, head towards opponent, hackles raised
Chase	Bird follows other bird with head high, other bird moving away
Attack	Bird moves swiftly towards opponent to give aggressive peck. Head over opponent
Perching	Sits or stands in any position on a perch
Zone sharing	Whether the bird is alone or with companions in its “zone” at the end of each 45 s of focal observation

## Data Availability

The complete dataset is available as Appendix A.

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
