# Peer review of "Selection for Reduced Fear of Humans Changes Intra-Specific Social Behavior in Red Junglefowl—Implications for Chicken Domestication"

_genes, 2021, doi:10.3390/genes13010043_

Round 1

Reviewer 1 Report

I think this is an extremely interesting paper, with clear results, properly-analized data and well-written.

Author Response

Thank you so much for your feedback and for your comments. We are happy you found our manuscript interesting 

Reviewer 2 Report

The experiment compared lines selected from a common junglefowl population for fear responses to humans.

  1. l. 14 from or form?
  2. l. 157 built or build?
  3. Reference section requires editing for consistency.
  4. l. 67-68 "highly crowded" in what context? Husbandry varies from cages to free range. This sentence requires documentation as it may or may not reflect what occurs in a range of husbandry settings.
  5. l. 93 Were the chickens of the same age (days)? Months is too vague.
  6. Frequency of crow (L > H). What about duration? The terminal component is brief in junglefowl. With tameness, did this change?
  7. l. 230 What do you mean by large offspring?
  8. l. 247 Was the increased aggression because they no longer recognized each other?
  9. In methods. It is not clear (or I may have missed it) whether or not the pairings were from a common flock. That is, were they strangers or not?

Author Response

Response reviewer #2

Thank you very much for your helpful comments. We have corrected and revised the manuscript according to your suggestions. Specifically, we changed as followed:

  1.Line 14, 14 from or form?

Answer: from; this has been changed

   2.Line 157, 157 built or build?

Answer: built; this has been changed.

   3.Reference section requires editing for consistency

Answer: Thank you for noticing, we have now edited the references section.

   4.Line 67-68, l. 67-68 "highly crowded" in what context? Husbandry varies from cages to free range. This sentence requires documentation as it may or may not reflect what occurs in a range of husbandry settings

Answer: l.68 -169 a new line is added with a new reference, documenting the density of broiler chickens in the EU as an example of “highly crowded”.

   5.Line 93, Were the chickens of the same age (days)? Months is too vague

Answer:  same age with max 11 days apart, see l.95.

   6.Frequency of crow (L > H). What about duration? The terminal component is  brief in junglefowl. With tameness, did this change?

Answer: A very interesting idea, but unfortunately, we did not record duration, only counts. This is clarified now in the method, l.144 and in the discussion, l.272 -273

  1. Line 230, 230 What do you mean by large offspring?

Answer: We added information and rephrased the sentence in l.230, so it is clear that it is the size of the chick that is larger not the number of chicks. We also added that the weight difference is about 10%.

  1. Line 247, Was the increased aggression because they no longer recognized each other?

Answer: We argue that that’s not the case since they are housed together when they are not in any experiment (separated by sex, not selection line) and they get two days of habituation with each other and the new enclosure before observation starts. We have now added this suggestion in l.253 -256

9.Methods, In methods. It is not clear (or I may have missed it) whether or not the pairings were from a common flock. That is, were they strangers or not,

Answer: They were not strangers, l.102 is rewritten so it will be clearer about the background of the chickens. Thank you for detecting this ambiguity.